# Selective Copolymerization from Mixed Monomers of Phthalic Anhydride, Propylene Oxide and Lactide Using Nano-Sized Zinc Glutarate

**DOI:** 10.3390/nano14181535

**Published:** 2024-09-22

**Authors:** Xiaoting Zhang, Zhidong Li, Liyan Wang, Jingjing Yu, Yefan Liu, Pengfei Song

**Affiliations:** College of Chemistry and Chemical Engineering, Key Laboratory of Eco-Functional Polymer Materials of the Ministry of Education, Key Laboratory of Eco-Environmental Polymer Materials of Gansu Province, Gansu International Scientific and Technological Cooperation Base of Water-Retention Chemical Functional Materials, Northwest Normal University, Lanzhou 730070, China

**Keywords:** selective polymerization, mixed monomers, ring-opening polymerization, zinc glutarate, block polyester

## Abstract

Selective polymerization with heterogeneous catalysts from mixed monomers remains a challenge in polymer synthesis. Herein, we describe that nano-sized zinc glutarate (ZnGA) can serve as a catalyst for the selective copolymerization of phthalic anhydride (PA), propylene oxide (PO) and lactide (LA). It was found that the ring-opening copolymerization (ROCOP) of PA with PO occurs firstly in the multicomponent polymerization. After the complete consumption of PA, the ring-opening polymerization (ROP) of LA turns into the formation of block polyester. In the process, the formation of zinc–alkoxide bonds on the surface of ZnGA accounts for the selective copolymerization from ROCOP to ROP. These results facilitate the understanding of the heterogeneous catalytic process and offer a new platform for selective polymerization from monomer mixtures.

## 1. Introduction

Selective copolymerization from mixed monomers can simplify process chemistry and eliminate the energy, time and labor currently used in intermediate separation, purification and protection–deprotection reaction steps [1,2]. It is superior to the traditional method for block polymers based on the sequential addition of monomers, pre-synthesis of macroinitiators, end-to-end coupling of pre-formed polymer chains and other two-step or multi-step routes [3,4,5,6]. Kinetic control has long been of interest as a means of controlling polymer compositions from mixtures, but this requires a common polymerization method [7]. Another strategy for the one-step formation of block polymers is selective copolymerization, which mainly depends on the nature of the catalyst with a combination of different catalytic cycles in a mixed-monomer system [8,9]. Undoubtedly, selective copolymerization from monomer mixtures is one of the most attractive means to prepare block copolymers. It is necessary to develop versatile catalysts that can be made available for these promising processes [10,11,12].

Polyesters are currently an important class of polymers that have been extensively investigated in the fields of biomedicine, packaging and engineering [13,14]. The synthesis of polyesters is generally achieved by the condensation polymerization of diols with diacids or diester, the ring-opening copolymerization (ROCOP) of epoxides with cyclic anhydrides and the ring-opening polymerization (ROP) of lactones [15,16,17,18]. Condensation polymerization for polyester synthesis is generally achieved at high temperatures to obtain polymers with a high molecular weight. In comparison, ROCOP and ROP can be performed in a controlled manner under mild conditions, producing desirable polyesters with excellent biocompatibility and mechanical properties [19,20]. A variety of catalysts including metal complexes [21,22,23,24,25,26,27,28] and organocatalysts [29,30,31,32,33,34,35,36,37] have been developed for ROCOP and ROP, respectively. This is of great interest regarding the rational design of catalysts applied for both ROCOP and ROP, which facilitate the combination of these two different catalytic cycles, leading to selective copolymerization from mixed monomers to synthesize block polyesters [38,39,40].

Generally, selective copolymerization with the combination of ROCOP and ROP depends on the functionality of catalysts and can switch between different polymerizations for sequential monomer addition. Accordingly, organometallic catalysts have been investigated for this promising process [41,42]. For example, Williams et al. prepared triblock polyesters by switching the catalytic combination of phthalic anhydride, vinyl epoxy cyclohexene and ε-decalactone using organometallic heterodinuclear Al (III)/K(I) complexes [23]. This catalyst could also be used to prepare multiblock polyesters and control the polymer structure. Zhu’s group developed a TU/PPNCl binary catalytic system that can switch between cyclic anhydride, epoxide and cyclic ester to prepare sequence-controlled multiblock polyesters with multiple structures. This method shows great potential for the synthesis of complex polymers with sequential and structural diversity [43]. Recently, Mazzeo et al. prepared chromium and aluminum complexes with a Salen ligand to obtain diblock polyesters (poly(propylene maleate-*block*-polyglycolide)) with precise compositions by switching catalytic reactions [44]. In fact, the controlled copolymerization of hybrid monomers often confers significant advantages on individual polymer chains and requires a single, multi-purpose catalyst, which remains a key challenge in the field of polymer chemistry [45]. Zinc glutarate (ZnGA) is a non-toxic catalyst that is easy to prepare and handle that has been investigated for the copolymerization of CO_2_ and epoxides producing aliphatic polycarbonate [46]. However, the heterogeneous catalytic action of ZnGA is still not clearly understood, which has hampered its application for the sequence-controlled copolymerization of mixed monomers. To achieve selective copolymerization using ZnGA, PA/PO ROCOP, LA ROP and the copolymerization of mixed monomers were investigated in detail (Figure 1).

## 2. Materials and Methods

### 2.1. Materials

Glutaric acid (GA), zinc oxide (ZnO) and L-lactide (LA, 99%) were used without further purification from Energy Chemical (Anhui, China). Propylene oxide (PO) was purchased from Energy Chemical and stored in an argon atmosphere in a vessel containing a pre-dried 3 Å molecular sieve. Phthalic anhydride (PA, AR, 99%) was purchased from Machlin Chemical (Shuanghai, China). Toluene, chloroform, n-hexane and anhydrous methanol were obtained from Sinopharm Chemical Reagent Co., Ltd. (Beijing, China). Toluene was stored in an argon atmosphere in a vessel containing a pre-dried 3 Å molecular sieve. Hydrochloric acid (36–38%) was procured from Beijing Chemical Works (Beijing, China).

### 2.2. Preparation of ZnGA Catalyst

The preparation of the nano-sized ZnGA catalyst was based on previous reports [47,48]. Glutaric acid (12.947 g, 98.0 mmol, 99.0%) and anhydrous toluene (150 mL) were added to a 250 mL three-necked flask, purged three times with nitrogen and kept under magnetic stirring; then, zinc oxide powder (8.139 g, 100.0 mmol, 99.8%) was added to the flask and refluxed for 8 h at 55 °C with magnetic stirring. The resulting solid was filtered and washed three times with acetone, and then the product was washed with a sodium chloride aqueous solution until the solution was neutral. The obtained ZnGA powder was dried under vacuum at 80 °C for 24 h, and the yield was calculated (yield = 97.6%).

### 2.3. The ROCOP of PA- and PO-Catalyzed ZnGA

The polymerization procedure is as follows: 0.2 g ZnGA was added to the reaction bottle and dried in a 120 °C vacuum oven overnight. After cooling to room temperature, PA (6.75 mmol, 1 equiv.) was added to the polymerization bottle, and the reaction bottle was sealed and vacuumed for 30 min. After that, PO (13.5 mmol, 2 equiv.) and 5 mL of toluene were sequentially added to the reaction bottle with a special syringe. Then, the polymerization was left to proceed at 120 °C for 4 h. To determine monomer conversion, a crude aliquot was time-regularly withdrawn from the polymerization by pipette and monitored by ^1^H NMR spectroscopy. After the defined time, the resulting mixture was diluted with 5% dilute hydrochloric acid and 3 mL chloroform, the polymer was washed with warm water for removing unreacted PA, and then it was precipitated with a large amount of n-hexane. Finally, the recovered solid was dried to a constant weight under vacuum and characterized by ^1^H NMR and GPC.

### 2.4. The ROP of LA Catalyzed by ZnGA/PO

The polymerization procedure is as follows: 0.2 g ZnGA was added to the reaction bottle and dried in a 120 °C vacuum oven overnight. After cooling to room temperature, LA (6.75 mmol, 1 equiv.) was added to the polymerization bottle, and the reaction bottle was sealed and vacuumed for 30 min. After that, PO (13.5 mmol, 2 equiv.) and 5 mL of toluene were sequentially added to the reaction bottle with a special syringe. Then, the polymerization was left to proceed at 120 °C for 4 h. To determine monomer conversion, a crude aliquot was time-regularly withdrawn from the polymerization by pipette and monitored by ^1^H NMR spectroscopy. After the defined time, the reaction mixture was diluted with approximately 5% dilute hydrochloric acid and 3 mL chloroform, and the polymer was precipitated with a large amount of methanol. Finally, the recovered solid was dried to a constant weight under vacuum and characterized by ^1^H NMR and GPC.

### 2.5. Copolymerization of PO-, PA- and LA-Catalyzed ZnGA

The polymerization procedure is as follows: 0.2 g ZnGA was added to the reaction bottle and dried in a 120 °C vacuum oven overnight. After cooling to room temperature, PA (6.75 mmol, 1 equiv.) and LA (6.75 mmol, 1 equiv.) were continuously added to the reaction bottle, and the reaction bottle was sealed and vacuumed for 30 min. After that, PO (13.5 mmol, 2 equiv.) and 5 mL of toluene were sequentially added to the reaction bottle with a special syringe. The molar feed ratio of PO, PA and LA was 2:1:1. Then, the polymerization was left to proceed at 120 °C for 9 h. To determine monomer conversion, a crude aliquot was time-regularly withdrawn from the polymerization by pipette and monitored by ^1^H NMR spectroscopy. After the defined time, the reaction mixture was diluted with 5% dilute hydrochloric acid and 3 mL chloroform, the polymer was washed by warm water for removing unreacted PA, and then it was precipitated with a large amount of methanol. Finally, the recovered solid was dried to a constant weight under vacuum and characterized by ^1^H NMR, GPC and DOSY NMR.

### 2.6. Characterization Methods

A magnetic test was conducted on a Brucker AM 600 M superconducting nuclear magnetic resonance (NMR) apparatus (Varian, USA) utilizing CDCl_3_ as the solvent and TMS as the internal standard. Fourier transform infrared (FTIR) spectra were tested on a Digilab Merlin FTS-3000 (Digilab, USA) infrared spectrometer by 64 scans from 4000 to 500 cm^−1^ with a spectral resolution of 4.0 cm^−1^. A gel permeation chromatography (GPC) test was performed with the United States Waters GPC2000 high-temperature gel permeation chromatographer and THF as the eluent. DOSY NMR analyses were performed at a steady temperature of 25 °C with at least 16 gradient increments using the ledbpgp2s sequence. The X-ray diffraction (XRD) patterns were recorded using a Brucher D8 DISCOVER (Bruker AXS, Germany) with Cu Ka radiation (g = 0.154 nm, 40 kV and 40 mA). The data were collected between 10° and 60° with a scanning speed of 2° min^−1^. Brunauer–Emmett–Teller (BET) surface areas were investigated by NovaWin (Quantachrom, USA). The pore volume and pore radius of the samples were determined by the Barrett–Joyner–Halenda method. The morphology of ZnGA was characterized using an ULTRA Plus Zeiss field emission scanning electron microscope (SEM).

## 3. Results and Discussion

### 3.1. Preparation and Characterization of ZnGA Catalyst

The ZnGA catalyst was prepared by the reaction of ZnO and GA. The structure of ZnGA was characterized by FTIR. As shown in Figure 1a, the absence of a peak at 1697 cm^−1^ (C=O) and the presence of peaks at 1585 cm^−1^, 1536 cm^−1^ and 1405 cm^−1^ (COO^−^) shows the formation of a zinc–carboxylate bond from the carbonyl group of GA. The absorbance of a CH_2_ shear band at 1443 cm^−1^ and CH stretching bonds at 2952 cm^−1^ were also detected, which indicates the successful preparation of the ZnGA catalyst. The degree of crystallinity of the ZnGA catalyst was characterized by XRD. As illustrated in the XRD plots, the ZnGA catalyst exhibit a distinct crystal structure (Figure 1b). To further illustrate the distinction between the diffraction peaks, the characteristic diffraction peaks with 2θ angles of 12.70°, 22.54° and 23.00° were selected, and the average grain sizes of the diffraction peaks were calculated to be below 60 nm by employing the Scherrer equation (Appendix A). In addition to crystallinity, the specific surface area is also the main factor affecting the catalytic activity of the catalyst. The specific surface area of the catalyst was further characterized by BET. The BET isotherm analysis of ZnGA shows that the catalyst is a macroporous material with a mean pore size of 60.3 nm and a surface area of 1.15 m^2^/g, which are typical values for ZnGA frameworks (Appendix A, Appendix A). In addition, the morphology of the ZnGA catalyst was characterized by SEM. As shown in Figure 1c,d, ZnGA showed a uniform nanosheet structure, providing the basis for good catalytic activity.

### 3.2. Selective Copolymerization of PA, PO and LA Using ZnGA

Firstly, the ROP of LA and ROCOP of PA with PO were carried out. Then, the copolymerization of PA, PO and LA was developed to combine ROCOP and ROP to achieve selective copolymerization. It is demonstrated that ZnGA can enable a selective conversion from monomer mixtures to synthesize block polyester. Initially, the ROCOP of PA and PO was performed with ZnGA under different conditions, and the copolymerization results are summarized in Table 1 and Appendix A. The results show that the ROCOP reaction is consistent with zero-order reaction kinetics, and the apparent rate constants (k_app_) were calculated to be 0.256 mol L^−1^ min^−1^. The GPC results showed that the molecular weight of PPAPO-co-PPO increased with the increase in the reaction time (Figure 2a).

As shown in Figure 2b, a new absorption peak at 1728 cm^−1^ appeared in the FTIR spectrum of the mixtures of ZnGA and PO, which indicates the formation of zinc–alkoxide species in the system [6]. It is considered that the zinc–alkoxide bond on the surface of the ZnGA can enable the ring-opening of PA and PO, which accounts for the ROCOP. As shown in Table 1 (entries 5–10), the PA conversion increased with the increase in the reaction temperature and the addition of PO (Appendix A). This is owing to the high temperature and high concentration of PO, which facilitates the reaction between ZnGA and PO to form zinc–alkoxide species. The polymerization of PO catalyzed by ZnGA was also investigated (Table 1, entry 11). It was found that the PO conversion was only 14% after 2 h, suggesting that the insertion of PO into the zinc–alkoxide bond is lower than that of PA. Accordingly, it is considered that the zinc–alkoxide bond on the surface of the ZnGA is active for the ROCOP, and the polymer chain growth takes place via the PA insertion and nucleophilic attack of PO, producing the aliphatic polyester with polyether segments in the chain. Furthermore, the ROP of LA with ZnGA was investigated in detail. Firstly, the polymerization of LA and PO with a molar feed ratio of 1:2 was performed at 120 °C (Appendix A). The results show that the ROP reaction is consistent with first-order reaction kinetics, and the k_app_ was calculated to be 0.019 mol L^−1^ min^−1^. Moreover, the GPC curves of the resulted PLA revealed a single peak, and the molecular weight of PLA increased with increasing reaction time (Figure 2c).

To explore the ROP process with ZnGA, we conducted the polymerization of LA without the addition of PO. However, it is shown that there was no PLA produced in the system in the absence of PO (Table 2, entry 1). It is suggested that the LA polymerization by ZnGA requires PO as an initiator, which facilitates the formation of the zinc–alkoxide bond, which accounts for the ROP of LA. Thus, the PLA chain growth can be achieved via the insertion of LA into the zinc–alkoxide bond. It is noted that the polymerization rate of LA with the zinc–alkoxide bond is much higher than that of PO, producing PLA without polyether segments in the PLA chain (Table 2, entries 2–7). Moreover, the polymerization of LA and PO with different molar feed ratios was investigated (Table 2, entries 7–10, Appendix A). It is shown that the ROP of LA with a high concentration of PO can produce PLA with a few polyether segments in the chain (Table 2, entry 10), which is owing to the formation of more zinc–alkoxide bonds on the surface of the ZnGA facilitating the homopolymerization of PO.

It is demonstrated that ZnGA can efficiently catalyze both the ROCOP of PA with PO and the ROP of LA, which encourages us to cultivate a selective copolymerization from mixed monomers to synthesize block polyesters (Appendix A). Accordingly, the copolymerization of PA, PO and LA with ZnGA were investigated in detail. The copolymerization process was monitored by ^1^H NMR analysis of aliquots taken at regular intervals (Figure 2d). It is obvious that the peaks at 8.05~7.85 ppm attributed to PA decreased with the increase in reaction time, while the peaks at 7.75~7.40 ppm, 5.49~5.36 ppm and 4.46~4.30 ppm are assigned to PPAPO, which increased with the increasing reaction time. Meanwhile, the peak at 5.07~5.00 ppm of LA remains unchanged until 7 h. It is suggested that the ROCOP of PA and PO was firstly initiated in the mixed monomers. After 7 h of the complete of PA, the increasing peak at 5.24~5.12 ppm of PLA was observed, implying that the ROP of LA was turned on. The copolymerization results are summarized in Table 3. It is demonstrated that a selective copolymerization including the first ROCOP and then ROP was achieved from mixed monomers of PA, PO and LA using ZnGA.

The GPC measurements showed that the copolymer had a single distribution (Figure 3a). Meanwhile, the molecular weight of the copolymer increased with the increase in reaction time, corresponding to the selective conversion of the first ROCOP of PA/PO and then the LA ROP. To verify the topology and purity of the copolymer, DOSY NMR spectrum was used, as shown in Figure 3b. It is clear that the resulting copolymer showed a single coefficient, indicating that only one component was exhibited in the resulting copolymer.

### 3.3. Polymerization Mechanism

To further understand the catalytic pathway, the polymerization of the PO and LA molar feed ratio of 2/1 at 120 °C was investigated. For the second phase at 0.5 h, an additional portion of PA was fed into the polymerization system (Table 4). As evidenced by ^1^H NMR spectroscopy (Figure 4a and Appendix A), the addition of PA immediately terminated the propagation of PLA. Meanwhile, the ROCOP of PA with PO was turned on in the presence of LA. It is suggested that the ROCOP of PA and PO is kinetically and thermodynamically favored over the ROP of LA. Accordingly, the selective copolymerization mechanism of PO, PA and LA can be proposed as shown in Figure 4b. Firstly, ZnGA can activate PO to form a zinc–alkoxide bond (-Zn-OR), which facilitates the insertion of PA to form a metal carboxyl terminal, followed by the ring opening of PO [49,50]. It is noted that the insertion rate of PA and PO into the zinc–alkoxide bond has little difference, leading to a small amount of polyether segments in the polyester chain. After the complete conversion of PA, the polymerization was turned on with the insertion of LA into the zinc–alkoxide bond producing block polyesters.

## 4. Conclusions

In summary, we have described a selective copolymerization route using nano-sized ZnGA to prepare block polyesters from mixed monomers in a one-pot process. The heterogeneous catalyst ZnGA can bridge two different catalytic polymerizations involving the ROCOP of PA/PO and the ROP of LA. In this switchable copolymerization, the LA ROP cannot proceed until PA is fully consumed in the multicomponent system, which mainly depends on the selectivity of the zinc–alkoxide bonds on the surface of the ZnGA for monomer mixture. This work offers the potential to develop a heterogeneous catalyst for the selective copolymerization from mixed monomers.

## Data Availability

Data are contained within the article.

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
