# Peer review of "Selective Copolymerization from Mixed Monomers of Phthalic Anhydride, Propylene Oxide and Lactide Using Nano-Sized Zinc Glutarate"

_nanomaterials, 2024, doi:10.3390/nano14181535_

Round 1

Reviewer 1 Report

Comments and Suggestions for Authors

Song et al. report on nano-sized zinc glutarate as catalyst for the ring-opening polymerisation of lactide and the ring-opening copolymerisation of phthalic anhydride with propylene oxide. Finally they also report on the ROP-ROCOP combination of all three monomers. In principle this type of research is interesting since the obtained polyesters are versatile in their properties and mostly biodegradable.

However, the study suffers from several technical weaknesses which may be solved in a revision:

a) line 93: "the white precipitate obtained by washing with excess acetone was washed and filtered several times until the solution was neutral..." -> with which solvent has been washes? besides acetone.

b) only in Figure 4 kinetics are shown and they are reported rather superficially. Please report more kinetics and then in a semilogarithmic form to determine if a controlled mechanism is present.

c) all tables: how often has every data point been determined?

d) please report polymerisation constants for a better comparison with the literature of ROP and ROCOP.

e) the authors combine 2 eq. of PO with 1 eq. of LA. Why? their data show, that LA performs a homopolymerisation and the PO forms the end groups.

f) the zinc catalyst is rather a heterogeneous catalyst: does the modification of the surface influence the kinetics of ROP and ROCOP?

Comments on the Quality of English Language

ok, but can be enhanced.

Author Response

Response to the Reviewers’ Comments

We appreciate the reviewers’ time evaluating our manuscript and their valuable comments and suggestions. The point-by-point answers to the concerns are provided below:

Response to Reviewer 1:

Comments:

Song et al. report on nano-sized zinc glutarate as catalyst for the ring-opening polymerisation of lactide and the ring-opening copolymerisation of phthalic anhydride with propylene oxide. Finally they also report on the ROP-ROCOP combination of all three monomers. In principle this type of research is interesting since the obtained polyesters are versatile in their properties and mostly biodegradable.

However, the study suffers from several technical weaknesses which may be solved in a revision:

a)     line 93: "the white precipitate obtained by washing with excess acetone was washed and filtered several times until the solution was neutral..." -> with which solvent has been washes? besides acetone.

Response: Thanks for your valuable comments. We have revised in the text. In the preparation of ZnGA catalyst, excess zinc oxide is added to ensure the complete reaction of glutaric acid. After the reaction, the resulting solids are filtered and washed with acetone three times to remove the toluene solvent. In order to improve the purity of ZnGA, the product is washed with sodium chloride solution until the solution is neutral.

b)     only in Figure 4 kinetics are shown and they are reported rather superficially. Please report more kinetics and then in a semilogarithmic form to determine if a controlled mechanism is present.

Response: Thanks for your valuable comments. Accordingly, the ROCOP kinetics of PA and PO (Figure S4, Page S-5 in the Supporting Information), as well as the ROP kinetics of LA (Figure S12, Page S-9 in the Supporting Information) was investigated in detail. It can be seen that the catalytic polymerization system with ZnGA has a controlled mechanism. 

c)     all tables: how often has every data point been determined?

Response: Thanks for your reminder. Every data point in all tables is carefully considered in order to study the controllability and mechanism of each reaction. 

d)     please report polymerisation constants for a better comparison with the literature of ROP and ROCOP.

Response: As shown in Table1 and Table2, the complete conversion of PA in ROCOP and LA in ROP in 4h, which are comparable to the ROCOP and ROP rate reported in the literature.

References

1)Wu, X.; Li, Y.; Yu, J.; Liu, Y.; Li, Z.; Zhang, Y.; Song, P., Switchable copolymerization of mixed monomers catalyzed by imidazolium ionic liquids. Polym. Chem. 2024, 15 (15), 1475-1483.

e)     the authors combine 2 eq. of PO with 1 eq. of LA. Why? their data show, that LA performs a homopolymerisation and the PO forms the end groups.

Response: Thanks for your valuable comments. In the polymerization of LA, zinc alkoxide bond (-Zn-OR) from ZnGA with PO is active for the LA ROP. Therefore, excess PO can form more active sites for LA ROP (Table2, entries 7-9, Figure S14, S15, line 211). It is suggested that PLA with end groups of PO can be prepared from this catalytic system.

f)     the zinc catalyst is rather a heterogeneous catalyst: does the modification of the surface influence the kinetics of ROP and ROCOP

Response: Thanks for your valuable comments. ZnGA powder surface contributes to the catalytic activity, so the surface modification of ZnGA has an influence on the kinetics of ROP and ROCOP. It is necessary to investigate the catalytic activity of modified ZnGA for the selective polymerization of mixed monomers in the future work. 

References

1)Yang, Y.; Lee, J. D.; Seo, Y. H.; Chae, J.-H.; Bang, S.; Cheong, Y.-J.; Lee, B. Y.; Lee, I.-H.; Son, S. U.; Jang, H.-Y., Surface activated zinc-glutarate for the copolymerization of CO2 and epoxides. Dalton Trans. 2022, 51 (43), 16620-16627.

2)Meng, Y. Z.; Du, L. C.; Tiong, S. C.; Zhu, Q.; Hay, A. S., Effects of the structure and morphology of zinc glutarate on the fixation of carbon dioxide into polymer. J. Polym. Sci. A Polym. Chem. 2002, 40 (21), 3579-3591. 

Response to Reviewer 2:

Comments:

Song reported the synthesis of nano-sized zinc glutarate and its application for copolymerization of phthalic anhydride, propylene oxide, and LA. LA ROP. The entire research is solid and informative. The publication of this research results is helpful to the research field of ROP. But I have several questions that the author needs to answer:

1.  In Scheme 1, the coefficient of monomers should be added. For example, the coefficient of PA should be “m” and “m+x” for PO in ROCOP. In ROP, PO was shown as the precursor, but it disappeared in the final product. What is the role of PO? I think the authors should recheck Scheme 1.

Response: Thanks for your valuable comments. We have revised Scheme 1 in the text. In the ROP, PO was employed to form zinc alkoxide bond (-Zn-OR) for the LA ROP. Moreover, there has no polyether segments were found in the prepared PLA chain. 

References

1) Duan, R.; Hu, C.; Liu, Y.; Bian, X.; Pang, X.; Chen, X., In Situ Initiation of Epoxides: Activated Metal Salt Catalysts for Cyclic Ester Polymerization. Ind. Eng. Chem. Res. 2022, 61 (51), 18712-18719.

2.  For the synthesis of ZnGA catalyst, are there any literature reports on such a synthesis method? Although the authors used many methods, such as FTIR, XRD, and SEM, to identify ZnGA catalyst, they do not prove its purity. I'm curious about how much Zn oxide remains in the product. Such doubts also make me think of the next question. Is part of all polymerization reactions catalyzed by Zn oxide? To eliminate my doubts and strengthen the necessity of using ZnGA catalyst, I suggest that Zn oxide and glutaric acid be used as catalysts in the polymerization reactions in Tables 1-3 as a control group.

Response: Thanks for your comments. The preparation of nano-sized ZnGA catalyst was based on previous reports. Excessive zinc oxide is added during the preparation of the catalyst to ensure the complete conversion of glutaric acid. To purify ZnGA, the resulting solid was filtered and washed three times with acetone, and then the product was washed with sodium chloride aqueous solution until the solution was neutral. 

References

1)Meng, Y. Z.; Du, L. C.; Tiong, S. C.; Zhu, Q.; Hay, A. S., Effects of the structure and morphology of zinc glutarate on the fixation of carbon dioxide into polymer. J. Polym. Sci. A Polym. Chem. 2002, 40 (21), 3579-3591. 

2)Padmanaban S, Dharmalingam S, Yoon S. A Zn-MOF-Catalyzed Terpolymerization of Propylene Oxide, CO2, and β-butyrolactone [J]. Catalysts 2018, 8(9): 393. 

3.  In LA ROP, the authors mentioned that PO was used as an initiator. However, I can see the PO chain end of producing PLA in Figure S10, and it may be ascribed to the large molecular mass of PLA. I suggested that the authors synthesize small molecular mass PLA (MnGPC about 500-1000 kg/mol) and report the 1H NMR spectrum to prove that PO was the initiator. In addition, the Figure S10 title mentions the PLA is from entry 7 of Table S2. Is it Table S2 or Table 2?

Response: Thanks for your valuable comments. In the polymerization of LA, zinc alkoxide bond (-Zn-OR) from ZnGA with PO is responsible for the initiation of LA ROP. It is suggested that PLA with end groups of PO can be prepared from this catalytic system. The Figure S10 title mentions the PLA is from entry 7 of Table 2. We have revised in the Supporting Information.

4. In Figure S2, where are the peaks of the methyl group on the PPO chain? What are the functional groups on the copolymer next to peaks c and d (at 4.25 and 5.25 ppm)? In the title of Figure S2, is the copolymer from Table S1 or Table 1?

Response: Thanks for your valuable comments. The hydrogen proton peak of the methyl group on the PPO chain is at the “h” position (Figure S2). The peaks at 4.25 ppm and 5.25 ppm are corresponding to the hydrogen proton on the methylene and methine groups of PPAPO polyester chain, respectively. The title of Figure S2 is the copolymer from Table 1, we have revised in the Supporting Information.

5. In Figure 4b, my understanding after reading this manuscript is that the author believes that if PA is added after opening the ring of PO, carboxylate will be formed after opening the ring of PA. Because carboxylate is a relatively weak initiator, it will not open the ring of LA to form PLA. If my understanding is correct, then the copolymer (PPA-PPO-PLA) on the bottom right side should not have PLA connected to PPO, but should be connected to the left side of PPA and extend from the carboxylate.

Response: Thanks for your valuable comments. Carboxylate is a relatively weak initiator, it will not open the ring of LA to form PLA, which is proved in the competitive reaction (Table 4, Figure 4a). As shown in Figure 4b, we proposed the selective copolymerization pathways for the monomers of PA, PO and LA by ZnGA, after the complete conversion of PA, the polymerization was turned on with the insertion of LA into zinc-alkoxide bond producing block polyesters. Thus, the copolymer (PPA-PPO-PLA) on the bottom right side should be PPAPO linked to PLA.

References

1)Meng, Y. Z.; Du, L. C.; Tiong, S. C.; Zhu, Q.; Hay, A. S., Effects of the structure and morphology of zinc glutarate on the fixation of carbon dioxide into polymer. J. Polym. Sci. A Polym. Chem. 2002, 40 (21), 3579-3591. 

2)Duan, R.; Hu, C.; Liu, Y.; Bian, X.; Pang, X.; Chen, X., In Situ Initiation of Epoxides: Activated Metal Salt Catalysts for Cyclic Ester Polymerization. Ind. Eng. Chem. Res. 2022, 61 (51), 18712-18719.

Reviewer 2 Report

Comments and Suggestions for Authors

Song reported the synthesis of nano-sized zinc glutarate and its application for copolymerization of phthalic anhydride, propylene oxide, and LA. LA ROP. The entire research is solid and informative. The publication of this research results is helpful to the research field of ROP. But I have several questions that the author needs to answer:

1.      In Scheme 1, the coefficient of monomers should be added. For example, the coefficient of PA should be “m” and “m+x” for PO in ROCOP. In ROP, PO was shown as the precursor, but it disappeared in the final product. What is the role of PO? I think the authors should recheck Scheme 1.

2.      For the synthesis of ZnGA catalyst, are there any literature reports on such a synthesis method? Although the authors used many methods, such as FTIR, XRD, and SEM, to identify ZnGA catalyst, they do not prove its purity. I'm curious about how much Zn oxide remains in the product. Such doubts also make me think of the next question. Is part of all polymerization reactions catalyzed by Zn oxide? To eliminate my doubts and strengthen the necessity of using ZnGA catalyst, I suggest that Zn oxide and glutaric acid be used as catalysts in the polymerization reactions in Tables 1-3 as a control group.

3.      In LA ROP, the authors mentioned that PO was used as an initiator. However, I can see the PO chain end of producing PLA in Figure S10, and it may be ascribed to the large molecular mass of PLA. I suggested that the authors synthesize small molecular mass PLA (MnGPC about 500-1000 kg/mol) and report the 1H NMR spectrum to prove that PO was the initiator. In addition, the Figure S10 title mentions the PLA is from entry 7 of Table S2. Is it Table S2 or Table 2?

4.      In Figure S2, where are the peaks of the methyl group on the PPO chain? What are the functional groups on the copolymer next to peaks c and d (at 4.25 and 5.25 ppm)? In the title of Figure S2, is the copolymer from Table S1 or Table 1?

5.      In Figure 4b, my understanding after reading this manuscript is that the author believes that if PA is added after opening the ring of PO, carboxylate will be formed after opening the ring of PA. Because carboxylate is a relatively weak initiator, it will not open the ring of LA to form PLA. If my understanding is correct, then the copolymer (PPA-PPO-PLA) on the bottom right side should not have PLA connected to PPO, but should be connected to the left side of PPA and extend from the carboxylate.

Author Response

Response to the Reviewers’ Comments

We appreciate the reviewers’ time evaluating our manuscript and their valuable comments and suggestions. The point-by-point answers to the concerns are provided below:

Response to Reviewer 1:

Comments:

Song et al. report on nano-sized zinc glutarate as catalyst for the ring-opening polymerisation of lactide and the ring-opening copolymerisation of phthalic anhydride with propylene oxide. Finally they also report on the ROP-ROCOP combination of all three monomers. In principle this type of research is interesting since the obtained polyesters are versatile in their properties and mostly biodegradable.

However, the study suffers from several technical weaknesses which may be solved in a revision:

a) line 93: "the white precipitate obtained by washing with excess acetone was washed and filtered several times until the solution was neutral..." -> with which solvent has been washes? besides acetone.

Response: Thanks for your valuable comments. We have revised in the text. In the preparation of ZnGA catalyst, excess zinc oxide is added to ensure the complete reaction of glutaric acid. After the reaction, the resulting solids are filtered and washed with acetone three times to remove the toluene solvent. In order to improve the purity of ZnGA, the product is washed with sodium chloride solution until the solution is neutral.

b) only in Figure 4 kinetics are shown and they are reported rather superficially. Please report more kinetics and then in a semilogarithmic form to determine if a controlled mechanism is present.

Response: Thanks for your valuable comments. Accordingly, the ROCOP kinetics of PA and PO (Figure S4, Page S-5 in the Supporting Information), as well as the ROP kinetics of LA (Figure S12, Page S-9 in the Supporting Information) was investigated in detail. It can be seen that the catalytic polymerization system with ZnGA has a controlled mechanism.

c) all tables: how often has every data point been determined?

Response: Thanks for your reminder. Every data point in all tables is carefully considered in order to study the controllability and mechanism of each reaction.

d) please report polymerisation constants for a better comparison with the literature of ROP and ROCOP.

Response: As shown in Table1 and Table2, the complete conversion of PA in ROCOP and LA in ROP in 4h, which are comparable to the ROCOP and ROP rate reported in the literature.

References

1)Wu, X.; Li, Y.; Yu, J.; Liu, Y.; Li, Z.; Zhang, Y.; Song, P., Switchable copolymerization of mixed monomers catalyzed by imidazolium ionic liquids. Polym. Chem. 2024, 15 (15), 1475-1483.

e) the authors combine 2 eq. of PO with 1 eq. of LA. Why? their data show, that LA performs a homopolymerisation and the PO forms the end groups.

Response: Thanks for your valuable comments. In the polymerization of LA, zinc alkoxide bond (-Zn-OR) from ZnGA with PO is active for the LA ROP. Therefore, excess PO can form more active sites for LA ROP (Table2, entries 7-9, Figure S14, S15, line 211). It is suggested that PLA with end groups of PO can be prepared from this catalytic system.

f) the zinc catalyst is rather a heterogeneous catalyst: does the modification of the surface influence the kinetics of ROP and ROCOP

Response: Thanks for your valuable comments. ZnGA powder surface contributes to the catalytic activity, so the surface modification of ZnGA has an influence on the kinetics of ROP and ROCOP. It is necessary to investigate the catalytic activity of modified ZnGA for the selective polymerization of mixed monomers in the future work.

References

1)Yang, Y.; Lee, J. D.; Seo, Y. H.; Chae, J.-H.; Bang, S.; Cheong, Y.-J.; Lee, B. Y.; Lee, I.-H.; Son, S. U.; Jang, H.-Y., Surface activated zinc-glutarate for the copolymerization of CO2 and epoxides. Dalton Trans. 2022, 51 (43), 16620-16627.

2)Meng, Y. Z.; Du, L. C.; Tiong, S. C.; Zhu, Q.; Hay, A. S., Effects of the structure and morphology of zinc glutarate on the fixation of carbon dioxide into polymer. J. Polym. Sci. A Polym. Chem. 2002, 40 (21), 3579-3591.

Response to Reviewer 2:

Comments:

Song reported the synthesis of nano-sized zinc glutarate and its application for copolymerization of phthalic anhydride, propylene oxide, and LA. LA ROP. The entire research is solid and informative. The publication of this research results is helpful to the research field of ROP. But I have several questions that the author needs to answer:

1. In Scheme 1, the coefficient of monomers should be added. For example, the coefficient of PA should be “m” and “m+x” for PO in ROCOP. In ROP, PO was shown as the precursor, but it disappeared in the final product. What is the role of PO? I think the authors should recheck Scheme 1.

Response: Thanks for your valuable comments. We have revised Scheme 1 in the text. In the ROP, PO was employed to form zinc alkoxide bond (-Zn-OR) for the LA ROP. Moreover, there has no polyether segments were found in the prepared PLA chain.

2. For the synthesis of ZnGA catalyst, are there any literature reports on such a synthesis method?Although the authors used many methods, such as FTIR, XRD, and SEM, to identify ZnGA catalyst, they do not prove its purity. I'm curious about how much Zn oxide remains in the product. Such doubts also make me think of the next question. Is part of all polymerization reactions catalyzed by Zn oxide? To eliminate my doubts and strengthen the necessity of using ZnGA catalyst, I suggest that Zn oxide and glutaric acid be used as catalysts in the polymerization reactions in Tables 1-3 as a control group.

Response: Thanks for your comments. The preparation of nano-sized ZnGA catalyst was based on previous reports. Excessive zinc oxide is added during the preparation of the catalyst to ensure the complete conversion of glutaric acid. To purify ZnGA, the resulting solid was filtered and washed three times with acetone, and then the product was washed with sodium chloride aqueous solution until the solution was neutral.

References

1)Meng, Y. Z.; Du, L. C.; Tiong, S. C.; Zhu, Q.; Hay, A. S., Effects of the structure and morphology of zinc glutarate on the fixation of carbon dioxide into polymer. J. Polym. Sci. A Polym. Chem. 2002, 40 (21), 3579-3591.

2)Padmanaban S, Dharmalingam S, Yoon S. A Zn-MOF-Catalyzed Terpolymerization of Propylene Oxide, CO2, and β-butyrolactone [J]. Catalysts 2018, 8(9): 393.

3. In LA ROP, the authors mentioned that PO was used as an initiator. However, I can see the PO chain end of producing PLA in Figure S10, and it may be ascribed to the large molecular mass of PLA. I suggested that the authors synthesize small molecular mass PLA (MnGPC about 500-1000 kg/mol) and report the 1H NMR spectrum to prove that PO was the initiator. In addition, the Figure S10 title mentions the PLA is from entry 7 of Table S2. Is it Table S2 or Table 2?

Response: Thanks for your valuable comments. In the polymerization of LA, zinc alkoxide bond (-Zn-OR) from ZnGA with PO is responsible for the initiation of LA ROP. It is suggested that PLA with end groups of PO can be prepared from this catalytic system. The Figure S10 title mentions the PLA is from entry 7 of Table 2. We have revised in the Supporting Information.

4. In Figure S2, where are the peaks of the methyl group on the PPO chain? What are the functional groups on the copolymer next to peaks c and d (at 25 and 5.25 ppm)?In the title of Figure S2, is the copolymer from Table S1 or Table 1?

Response: Thanks for your valuable comments. The hydrogen proton peak of the methyl group on the PPO chain is at the “h” position (Figure S2). The peaks at 4.25 ppm and 5.25 ppm are corresponding to the hydrogen proton on the methylene and methine groups of PPAPO polyester chain, respectively. The title of Figure S2 is the copolymer from Table 1, we have revised in the Supporting Information.

5. In Figure 4b, my understanding after reading this manuscript is that the author believes that if PA is added after opening the ring of PO, carboxylate will be formed after opening the ring of PA.Because carboxylate is a relatively weak initiator, it will not open the ring of LA to form PLA. If my understanding is correct, then the copolymer (PPA-PPO-PLA) on the bottom right side should not have PLA connected to PPO, but should be connected to the left side of PPA and extend from the carboxylate.

Response: Thanks for your valuable comments. Carboxylate is a relatively weak initiator, it will not open the ring of LA to form PLA, which is proved in the competitive reaction (Table 4, Figure 4a). As shown in Figure 4b, we proposed the selective copolymerization pathways for the monomers of PA, PO and LA by ZnGA, after the complete conversion of PA, the polymerization was turned on with the insertion of LA into zinc-alkoxide bond producing block polyesters. Thus, the copolymer (PPA-PPO-PLA) on the bottom right side should be PPAPO linked to PLA.

References

1)Meng, Y. Z.; Du, L. C.; Tiong, S. C.; Zhu, Q.; Hay, A. S., Effects of the structure and morphology of zinc glutarate on the fixation of carbon dioxide into polymer. J. Polym. Sci. A Polym. Chem. 2002, 40 (21), 3579-3591.

2)Duan, R.; Hu, C.; Liu, Y.; Bian, X.; Pang, X.; Chen, X., In Situ Initiation of Epoxides: Activated Metal Salt Catalysts for Cyclic Ester Polymerization. Ind. Eng. Chem. Res. 2022, 61 (51), 18712-18719.

Round 2

Reviewer 1 Report

Comments and Suggestions for Authors

Song et al. provide a revised version of their manuscript. Many points have been addressed but some major points are still open:

c) all tables: how often has every data point been determined?
Response: Thanks for your reminder. Every data point in all tables is carefully
considered in order to study the controllability and mechanism of each reaction

My question more precise: how often has every measurement/kinetics been performed?

d) please report polymerisation constants for a better comparison with the literature
of ROP and ROCOP.
Response: As shown in Table1 and Table2, the complete conversion of PA in
ROCOP and LA in ROP in 4h, which are comparable to the ROCOP and ROP
rate reported in the literature.

My question more precise: 4 h is not a polymerisation constant. Please report rate constants such as k_app or k_obs.

e) the authors combine 2 eq. of PO with 1 eq. of LA. Why? their data show, that LA performs a homopolymerisation and the PO forms the end groups.
Response: Thanks for your valuable comments. In the polymerization of LA, zinc alkoxide bond (-Zn-OR) from ZnGA with PO is active for the LA ROP. Therefore, excess PO can form more active sites for LA ROP (Table2, entries 7-9, Figure S14, S15, line 211). It is suggested that PLA with end groups of PO can be prepared from this catalytic system.

My question more precise: I understand that the authors want to obtain PLA with PO end groups, but herefore, you do not need 2 eq of PO vs LA, but rather 0.2 eq. So please explain the strategy.

f) the zinc catalyst is rather a heterogeneous catalyst: does the modification of the surface influence the kinetics of ROP and ROCOP
Response: Thanks for your valuable comments. ZnGA powder surface
contributes to the catalytic activity, so the surface modification of ZnGA has an influence on the kinetics of ROP and ROCOP. It is necessary to investigate the catalytic activity of modified ZnGA for the selective polymerization of mixed monomers in the future work.

My question more precise: This does not answer my question. How does the surface modification of ZnGA influence the kinetics?

After re-revision, the manuscript will become suitable for publication.

Comments on the Quality of English Language

Mostly ok.

Author Response

Response to Reviewer 1:

Comments:

Song et al. provide a revised version of their manuscript. Many points have been addressed but some major points are still open:

c) all tables: how often has every data point been determined?

My question more precise: how often has every measurement/kinetics been performed?

Response: Thanks for your comments. Each data point in Tables and Figures is an independent copolymerization reaction under standard conditions. As shown in Table 1, half an hour difference in the each reaction time.

d) please report polymerisation constants for a better comparison with the literature

of ROP and ROCOP.

My question more precise: 4 h is not a polymerisation constant. Please report rate constants such as k_app or k_obs.

Response: Thanks for your valuable comments. The POCOP reaction of PA and PO catalyzed by ZnGA is consistent with zero-order reaction kinetics (Figure S4), and the ROP reaction of LA catalyzed by ZnGA is consistent with first-order reaction kinetics (Figure S12). The apparent copolymerization rate constants (kapp) were calculated to be 0.256 mol L-1 min-1 and 0.019 mol L-1 min-1, respectively. Corresponding changes have been made in the revised manuscript.

e) the authors combine 2 eq. of PO with 1 eq. of LA. Why? their data show, that LA performs a homopolymerisation and the PO forms the end groups.

My question more precise: I understand that the authors want to obtain PLA with PO end groups, but herefore, you do not need 2 eq of PO vs LA, but rather 0.2 eq. So please explain the strategy.

Response: Thanks for your valuable comments. As shown in Table 2, the moral ratio of PO/LA for LA ROP was investigated, indicating that 2eq of PO vs LA facilitates for the LA polymerization. It is considered that ZnGA with excess PO can form more active sites for LA ROP (Table 2, entries 7-9), including the copolymerization of PA, PO and LA to produce block polyesters (Table 3).

f) the zinc catalyst is rather a heterogeneous catalyst: does the modification of the surface influence the kinetics of ROP and ROCOP.

My question more precise: This does not answer my question. How does the surface modification of ZnGA influence the kinetics?

Response: Thanks for your valuable comments. The reasonable surface modification of ZnGA including the increase of specific surface area and surface basicity can improve the catalytic activity of ZnGA for ROP and ROCOP.

References

1) Yoon S. Surface modification of a MOF-based catalyst with lewis metal salts for improved catalytic activity in the fixation of CO2 into polymers[J]. Catalysts, 2019, 9: 892.

2) Yang Y, Lee J D, Seo Y H, et al. Surface activated zinc-glutarate for the copolymerization of CO2 and epoxides[J]. Dalton Transactions, 2022, 51(43): 16620-16627.

Reviewer 2 Report

Comments and Suggestions for Authors

I am satisfied with the author's reply. This manuscript is recommended for publication

Author Response

Response to Reviewer 2:

Comments:

I am satisfied with the author's reply. This manuscript is recommended for publication.

Response: Thanks for your valuable comments.

Round 3

Reviewer 1 Report

Comments and Suggestions for Authors

Song et al. provide a re-revised version of their manuscript. Many points have been addressed but some major points are still open:

c) all tables: how often has every data point been determined?

My question more precise: how often has every measurement/kinetics been performed?

Response: Thanks for your comments. Each data point in Tables and Figures is an independent copolymerization reaction under standard conditions. As shown in Table 1, half an hour difference in the each reaction time.

My reply: the experiments should be performed at least twice to check for reproducibility.

e) the authors combine 2 eq. of PO with 1 eq. of LA. Why? their data show, that LA performs a homopolymerisation and the PO forms the end groups.

My question more precise: I understand that the authors want to obtain PLA with PO end groups, but herefore, you do not need 2 eq of PO vs LA, but rather 0.2 eq. So please explain the strategy.

Response: Thanks for your valuable comments. As shown in Table 2, the moral ratio of PO/LA for LA ROP was investigated, indicating that 2eq of PO vs LA facilitates for the LA polymerization. It is considered that ZnGA with excess PO can form more active sites for LA ROP (Table 2, entries 7-9), including the copolymerization of PA, PO and LA to produce block polyesters (Table 3).

My reply: Table 2 shows only few ratios, such as 0/1, 1/1, 2/1 (and not very rationally 2/0.5). It should be also tested like 0.1/1 and 0.2/1 to have meaningful results.

 Further point: the authors state "Data Availability Statement: Data are contained within the article. "

-> This is not correct. 

The article may be suited after revision.

Author Response

Response to Reviewer 1:

Song et al. provide a re-revised version of their manuscript. Many points have been addressed but some major points are still open:

c) all tables: how often has every data point been determined?

My question more precise: how often has every measurement/kinetics been performed?

Response: Thanks for your comments. Each data point in Tables and Figures is an independent copolymerization reaction under standard conditions. As shown in Table 1, half an hour difference in the each reaction time.

My reply: the experiments should be performed at least twice to check for reproducibility.

Response: Thanks for your kind suggestion. Each data point in all tables and figures has been repeated to ensure accuracy.

e) the authors combine 2 eq. of PO with 1 eq. of LA. Why? their data show, that LA performs a homopolymerisation and the PO forms the end groups.

My question more precise: I understand that the authors want to obtain PLA with PO end groups, but herefore, you do not need 2 eq of PO vs LA, but rather 0.2 eq. So please explain the strategy.

Response: Thanks for your valuable comments. As shown in Table 2, the moral ratio of PO/LA for LA ROP was investigated, indicating that 2eq of PO vs LA facilitates for the LA polymerization. It is considered that ZnGA with excess PO can form more active sites for LA ROP (Table 2, entries 7-9), including the copolymerization of PA, PO and LA to produce block polyesters (Table 3).

My reply: Table 2 shows only few ratios, such as 0/1, 1/1, 2/1 (and not very rationally 2/0.5). It should be also tested like 0.1/1 and 0.2/1 to have meaningful results.

Response: We appreciate your kind suggestion. The LA ROP with moral ratio of PO/LA=0.2/1 was investigated (Table 2, line 8, highlighted in red), the results showed that the conversion rate of LA was only 65% under the same conditions, indicating that 0.2 eq. of PO cannot form sufficient active sites for LA ROP. Therefore, it is considered that 2eq. of PO vs LA is more efficient for LA ROP, same results are reported in related literatures1,2.   

References

  1. Duan, R.; Hu, C.; Li, X.; Pang, X.; Sun, Z.; Chen, X.; Wang, X., Air-Stable Salen–Iron Complexes: Stereoselective Catalysts for Lactide and ε-Caprolactone Polymerization through in Situ Initiation. Macromolecules 2017, 50 (23), 9188-9195.
  2. Duan, R.; Hu, C.; Liu, Y.; Bian, X.; Pang, X.; Chen, X., In Situ Initiation of Epoxides: Activated Metal Salt Catalysts for Cyclic Ester Polymerization. Industrial & Engineering Chemistry Research 2022, 61 (51), 18712-18719.
